# Plasma Polyunsaturated Fatty Acid Levels and Mental Health in Middle-Aged and Elderly Adults

**DOI:** 10.3390/nu16234065

**Published:** 2024-11-26

**Authors:** Yongxuan Li, Li Hua, Qingqing Ran, Jiawei Gu, Yujia Bao, Jinli Sun, Lan Wu, Mu He, Yuzheng Zhang, Jinxin Gu, Jinjun Ran

**Affiliations:** 1School of Public Health, Shanghai Jiao Tong University School of Medicine, Shanghai 200025, China; melody321@sjtu.edu.cn (Y.L.); seyhuali@shsmu.edu.cn (L.H.); youkiq@sjtu.edu.cn (Q.R.); jw.gu0312@sjtu.edu.cn (J.G.); bubble-y@sjtu.edu.cn (Y.B.); sunjinli1989@shsmu.edu.cn (J.S.); 2School of Mathematics and Physics, Xi’an Jiaotong-Liverpool University, Suzhou 215123, China; roxannewu0419@gmail.com (L.W.); mu.he@xjtlu.edu.cn (M.H.); 3China National Health Development Research Centre, Beijing 100032, China; zhangyuzheng46@163.com; 4Academy of Pharmacy, Xi’an Jiaotong-Liverpool University, Suzhou 215123, China

**Keywords:** omega-3, omega-6, mental health, psychiatric symptoms, white matter microstructures

## Abstract

**Background:** Polyunsaturated fatty acids (PUFAs) are promising nutrients for the prevention and management of psychiatric disorders. Both animal experiments and cohort studies have demonstrated the antidepressant effects of PUFAs, especially omega-3 PUFAs. However, inconsistent reports about specific types of PUFAs, such as the omega-3 and omega-6 PUFAs, still exist. **Objectives:** To assess the effects of specific PUFAs on mental disorders and related symptoms and explore the potential mechanisms involving white matter microstructure. **Methods:** Leveraging 102,252 residents from the UK Biobank, the effects of five PUFA measures on depressive disorder and anxiety disorder were explored through Cox regression models with full adjustment for possible confounders. Furthermore, the effects on related psychiatric symptoms and brain white matter microstructures were also estimated using logistic regression models and multiple linear regression models, respectively. **Results:** In this study, plasma levels of five PUFAs measured in quartile 4 were associated with lower risks of incident depressive disorder compared with the lowest quartile, with hazard ratios of 0.80 [95% confidence interval] = [0.71, 0.90] for total PUFAs, 0.86 [0.76, 0.97] for omega-3 PUFAs, 0.80 [0.71, 0.91] for docosahexaenoic acid, 0.79 [0.70, 0.89] for omega-6 PUFAs, and 0.77 [0.69, 0.87] for linoleic acid. Similar associations were observed between PUFAs and the incident risk of anxiety disorder. In addition, high plasma PUFA levels were also related to lower risks of occurrence of several adverse psychological symptoms, especially omega-3 PUFAs and DHA. Among the included participants, 8780 individuals with brain imaging information were included in further neuroimaging analyses, and significant associations with white matter microstructures were observed. **Conclusions:** Thus, this study provides population-based evidence to support the value of interventions to target PUFAs (specifically omega-3 PUFAs) for the prevention and improvement of mental health.

## 1. Introduction

Mental health is considered a fundamental human right and essential to the development of all countries. The increasing prevalence of mental disorders, including depressive disorder and anxiety disorder, is a significant public health concern that imposes a considerable burden on society, families, and individuals alike [1]. With the population aging, statistical evidence suggests that around 14% of adults aged 60 and over live with a mental disorder, with an anticipated rise in this figure [2]. Despite extensive research efforts spanning decades, the precise etiology of these mental disorders remains elusive. Clinical and epidemiological studies are increasingly concentrated on the identification of diverse modifiable risk and protective factors, which are of paramount importance for the prevention of the occurrence and expeditious advancement of mental disorders. While social and behavioral epidemiology is sufficient, nutritional evidence is limited. In recent years, there has been a growing interest in the potential roles of specific nutrients, food, or dietary patterns in the precise intervention and prevention of mental disorders.

Polyunsaturated fatty acids (PUFAs) are mainly comprised of omega-3 (*n*-3) and omega-6 (*n*-6) fatty acids. While the brain can synthesize the majority of required saturated and monounsaturated fatty acids, it lacks the ability to synthesize essential PUFAs. The essential PUFAs, the “parent” of the omega-3 and omega-6 fatty acid families, must be supplied by diet. PUFAs, particularly DHA, are abundant in the brain and esterified in the membrane phospholipids of neurons and glial cells to regulate both structure and function [3]. The significance of PUFAs in neurodevelopment, aging, and neurodegeneration has been substantiated in both animal experiments and clinical studies, with a particular emphasis on omega-3 fatty acids, including docosahexaenoic acid (DHA) and eicosapentaenoic acid (EPA) [4,5,6]. Several animal experiments and epidemiology studies have reported that omega-3 fatty acids, especially EPA, have antidepressant effects that can be attributed to their modulation of neuroinflammation, neurotransmitter function, and neuroplasticity [7,8,9]. However, recent randomized controlled trials (RCTs) present mixed results, with some indicating that EPA-rich omega-3 supplements benefit patients with depression [10], while others find minimal or no significant effects [11]. Several studies have explored the roles of *n*-6 PUFAs in mental health but the epidemiological evidence remains scarce. Although limited studies have demonstrated that high omega-6 levels are related to an increased risk of depression [12,13], recent findings have highlighted that moderate omega-6 intake within a recommended level would be beneficial for mental health [14]. Of note, benefitting from the UK Biobank, there are numerous studies providing population evidence on the important roles of PUFAs in all-cause mortality, cardiovascular diseases, and dementia [15,16,17], while population-based studies on the associations with mental disorders remain scarce. To fill the gaps mentioned above, it is therefore imperative to investigate the relationship between specific types of PUFAs and mental health, as well as the involved brain imaging phenotypes based on a large population, in order to gain insights that can inform further research into disease pathogenesis and human brain health.

In this study, we assessed the relationships of plasma PUFA levels, including five measures, with depressive disorder and anxiety disorder, as well as the related psychiatric symptoms, leveraging a large prospective UK cohort. Furthermore, we explored the associations of plasma PUFA levels with white matter microstructures. Taken together, this study may support the value of interventions targeting PUFAs to improve mental health.

## 2. Materials and Methods

### 2.1. Study Population

The UK Biobank is a UK-based population cohort that recruited about half a million participants aged 37–73 years within 22 assessment centers across England, Scotland, and Wales. The baseline assessment was conducted between 2006 and 2010. The sociodemographic characteristics and performed anthropometric measurements were derived by a touchscreen questionnaire at baseline. After recruitment, the participants were followed up for mortality and incidence through electronic linkages with national health records. The dietary intake and imaging information were also collected during follow-up. The UK Biobank has been approved by the North West Multi-Center Research Ethics Committee (Reference: 11/NW/03820). The UK Biobank’s informed consent process and ethical approval are well-documented (https://www.ukbiobank.ac.uk/learn-more-about-uk-biobank/about-us/ethics (accessed on 3 October 2024)). This study was approved with the application number 99001, and the approval period is from 31 March 2023 to 31 January 2026. Among the project dataset that included 502,369 UK residents, we excluded participants with prevalent diseases at baseline, those without exposure information, and those without data for specific covariates. A total of 102,252 participants were included in this study, where 8780 individuals with data on brain imaging phenotypes were included in further analyses. This study followed the Strengthening the Reporting of Observational Studies in Epidemiology (STROBE) guidelines.

### 2.2. Plasma PUFA Levels

Metabolomic profiling of plasma samples was performed with high-throughput nuclear magnetic resonance (NMR) spectroscopy. Plasma PUFA levels were quantified in baseline blood samples using this method. Detailed protocols regarding the sample preparation, NMR setup, quality control, dilution correction, and the verification process through duplicate samples and internal controls, as well as comparisons with clinical chemistry assays, have been described in previous publications [18]. Five PUFA-related measures were derived in absolute concentration units (mmol/L), including total PUFAs, total omega-3 PUFAs, total omega-6 PUFAs, docosahexaenoic acid (DHA), and linoleic acid (LA). Of note, DHA is one type of omega-3 PUFA, and LA is one type of omega-6 PUFA. Given that individual omega-3 species other than DHA and omega-6 species other than LA are not available in the UK Biobank, to examine whether other PUFAs are associated with incident mental disorders, we further calculated “non-DHA omega-3 PUFAs” by subtracting the DHA value from the total omega-3 value and calculated “non-LA omega-6 PUFAs” by subtracting the LA value from the total omega-6 value.

### 2.3. Outcome

The primary outcomes included depressive disorder, anxiety disorder, and 19 psychiatric symptoms. Data on hospital inpatient diagnoses in the UK Biobank repository were accessed through linkage to national registers. Cases were coded using the International Classification of Diseases 10th edition (ICD-10): major depressive disorder (F32 and F33) and anxiety disorder (F40 and F41). All participants were followed from the date of attending the baseline assessment center to the endpoints, including the earliest date of any brain disorder diagnosis, death, withdrawal from the UK Biobank cohort, the last available date from either primary care data (UKB field 42040) or hospital inpatient data (UKB field 41280–41281), or 19 December 2022, whichever occurred first. Psychiatric symptoms were gathered from the UK Biobank’s online mental health self-assessment questionnaire (category 136). Due to the high level of missing data for certain items, 19 symptoms with a large sample were selected, representing three key groups: subjective well-being, depressive symptoms, and anxiety symptoms. Specifically, subjective well-being was measured through three questions asking participants how happy or satisfied they felt regarding their happiness, health, and life. Individuals who selected “Extremely unhappy” or “Not at all” were classified as experiencing unhappiness. Depressive and anxiety symptoms were evaluated through the Patient Health Questionnaire 9 (PHQ-9) and Generalized Anxiety Disorder (GAD-7), respectively. These questionnaires used a 4-point Likert scale, ranging from 0 (“not at all”) to 3 (“nearly every day”). A score of 1 or higher on each item indicated the presence of the corresponding symptom. The detailed information for the definition of psychiatric symptoms in the UK Biobank is presented in Appendix A.

The secondary outcomes included multiple brain white matter tracts assessed by indicators including mean diffusivity (MD) and isotropic volume fraction (ISOVF) values derived from the water diffusion magnetic resonance imaging (dMRI). MD reflects the average magnitude of water diffusion within a tissue, offering a measure of overall diffusivity across all directions. This metric provides an insight into the tissue microstructure, with increased MD values often indicating a loss of microstructural integrity. Diffusion tensor imaging (DTI) was used to compute MD, where water diffusion was modeled in multiple directions, and the average of the three eigenvalues from the diffusion tensor was taken to represent MD. ISOVF, on the other hand, is an indicator derived from the NODDI (Neurite Orientation Dispersion and Density Imaging) model. ISOVF measures the fraction of isotropic diffusion within a voxel, which typically reflects the volume of free water or cerebrospinal fluid (CSF). The NODDI model calculates ISOVF by fitting multi-shell diffusion data to estimate the proportion of water diffusion that is not constrained by cell membranes. Data collection for these measures followed the protocols established by the UK Biobank neuroimaging initiative. Participants underwent dMRI scans using a Siemens Skyra 3 T scanner, with diffusion gradients applied in 100 directions and *b*-values of 1000 s/mm^2^, 2000 s/mm^2^, and 3000 s/mm^2^. The images were preprocessed, including motion correction, eddy current correction, and distortion correction. The resulting diffusion-weighted images were then used to fit the diffusion tensor model for calculating MD and the NODDI model for estimating ISOVF. Among the 102,252 participants, 8780 individuals with brain imaging information were included to conduct further neuroimaging analysis. All continuous indices of brain structures were standardized using Z-transformation for interpretation and comparison. For detailed information on white matter tracts in the UK Biobank, refer to Appendix A.

### 2.4. Covariates

To control the potential bias arising from confounders related to psychiatric disorders or plasma PUFA levels, we considered age, sex, the index of multiple deprivation (IMD), waist-to-hip ratio (WHR), no current smoking, a healthy sleep pattern, healthy diet, regular physical activity, systolic blood pressure, diastolic blood pressure, and blood glucose levels as covariates in the statistical models based on previous evidence. The WHR was calculated as the waist circumference divided by the hip circumference. The IMD was used to measure social deprivation from the neighborhood level, which consists of 7 domains involving health, education, and others. In terms of healthy lifestyles, the smoking status was categorized as never smoked or current smoking. A healthy sleep pattern was defined based on at least four of five aspects of healthy sleep behaviors. The components were as follows: early chronotype; sleep 7–8 h/day; never/rarely or sometimes insomnia; no self-reported snoring; and never/rarely or sometimes daytime dozing [19,20]. A healthy diet was defined based on at least four of seven aspects of healthy diet components following recommendations on dietary priorities for cardiometabolic health [21]. The components were as follows: fruits ≥ 3 servings/day; vegetables ≥ 3 servings/day; fish ≥ 2 servings/week; processed meats ≤ 1 serving/week; unprocessed red meats ≤ 1.5 servings/week; whole grains ≥ 3 servings/day; and refined grains ≤ 1.5 servings/day. Regular physical activity was defined as meeting at least 150 min of moderate activity per week, 75 min of vigorous activity, or an equivalent combination [22]. The inflammatory condition (low-grade inflammation measured by the INFLA score) and metabolic syndrome (MetS) were also considered in the subgroup analysis. The definitions and calculations could be found in previous studies [23,24].

### 2.5. Statistical Analysis

The baseline characteristics of the study participants were summarized as means (standard deviations [SDs]) for continuous variables and counts (percentages) for categorical variables. Five PUFA-related measures were considered as exposures, including total PUFAs, total omega-3 PUFAs, total omega-6 PUFAs, DHA, and LA. After ensuring that the proportional hazard assumption was met through the Scaled Schoenfeld Residual, Cox proportional hazard regression models were utilized to assess the relationships between exposures of interest and the incident risks of mental disorders with quartiles (with Q1 as the reference) and quintiles (with the lowest quintile as the reference). Missing values were deleted from the analyses. Three model strategies were adopted: Model 1 was adjusted for baseline characteristics, including age, sex, WHR, and IMD; Model 2 was additionally adjusted for healthy lifestyles (no current smoking, a healthy sleep pattern, healthy diet, and regular physical activity); while Model 3 was further additionally adjusted for systolic blood pressure, diastolic blood pressure, and blood glucose levels. Hazard ratios (HRs) and corresponding 95% confidence intervals (CIs) for plasma PUFA levels in each quartile or quintile, with the corresponding Q1 or lowest quintile as a reference, were calculated using Cox proportional hazards regression models. HRs and corresponding 95% CIs per interquartile range (IQR) were also estimated. To further identify the potential modifiers of the associations above, a series of stratified analyses were performed by age (<60/≥60), sex (male/female), INFLA score (measurement of low-grade inflammation [low/high]), and MetS (yes/no). A test of the interaction between the plasma PUFA level and each term was conducted in the model by including a multiplicative term.

The effects of plasma PUFA levels in quartile 4 (with Q1 as a reference) on 19 psychiatric symptoms were further investigated with the logistic regression model using the same adjustment strategy. The results are estimated as odds ratios (ORs) with their corresponding 95% CIs. For neuroimaging analyses, a multiple linear regression model was utilized to estimate the associations between the plasma PUFA levels and white matter microstructures (MD and ISOVF indicators), with the same adjustment as the fully adjusted model mentioned above. Effect values were shown as the *β*s with corresponding 95% CIs. An analysis stratified by age (<60/≥60) and sex (female/male) was conducted to assess the potential effect modification. False discovery rate (FDR) control was used to compensate for multiple comparisons for all *p* values.

Several sensitivity analyses were conducted to evaluate the robustness of our results. First, we restricted the cohort to individuals with White European ancestry, given that non-White participants represent only approximately 3% of the total dataset. Second, participants who had been diagnosed with a targeted disease event within the first two years of follow-up were excluded from the study. Third, in consideration of the impact of the COVID-19 outbreak on our findings, we advanced the deadline for follow-up to 31 December 2019 [25]. Fourth, we further adjusted for no heavy alcohol intake in the model, which could also be a potential confounder in the associations above. Finally, considering the potential effects of plasma lipid metabolites, we further adjusted for the plasma cholesterol and triglyceride levels [26]. Statistical significance was determined by a two-tailed *p*-value of less than 0.05. All analyses were conducted using R software, version 4.2.2.

## 3. Results

Among the project, the dataset included 502,369 UK residents, and a total of 102,252 participants were enrolled in this study following the exclusion of 400,117 participants lacking data on plasma PUFA levels and information on potential confounding variables, as well as those diagnosed with a specific outcome at baseline (Figure 1). Among the participants, 56.6% were under the age of 60, 53.2% were female, 50.0% had a high IMD, and 48.7% had a poor WHR. Statistics of the baseline information for the individuals across quartiles of plasma total PUFA levels are shown in Table 1. During the follow-up time, 3411 (3.34%) participants were diagnosed with depressive disorder and 3326 (3.25%) with anxiety disorder. Appendix A presents data on the occurrence of depressive disorder and anxiety disorder, expressed in terms of total person-years and incidence rates, disaggregated by sex.

After full adjustment for age, sex, IMD, WHR, a healthy lifestyle (no current smoking, healthy sleep pattern, healthy diet, and regular physical activity), systolic blood pressure, diastolic blood pressure, and blood glucose levels, the highest quartiles of the five PUFA measures were significantly associated with a lower incident risk of depressive disorder compared with the lowest quartiles, with HRs of 0.80 [95% CI] = [0.71, 0.90] for total PUFAs, 0.86 [0.76, 0.97] for omega-3 PUFAs, 0.80 [0.71, 0.91] for DHA, 0.79 [0.70, 0.89] for omega-6 PUFAs, and 0.77 [0.69, 0.87] for LA (Table 2). Similar relationships were also observed for the highest quartiles of the five PUFA measures with a lower incident risk of anxiety disorder, with 0.83 [0.74, 0.94] for total PUFAs, 0.83 [0.73, 0.94] for omega-3 PUFAs, 0.80 [0.71, 0.91] for DHA, 0.84 [0.75, 0.96] for omega-6 PUFAs, and 0.84 [0.75, 0.95] for LA. (Table 2). In addition, similar relationships of five PUFA measures with lower risks of mental disorders were also observed when the PUFA levels were divided into quintiles, with quintile 1 as reference (Table 3). Significant associations were also observed in the continuous analysis (per IQR). In addition, we also considered non-DHA omega-3 PUFAs and non-LA omega-6 PUFAs as exposures. For non-DHA omega-3 PUFAs, marginally significant associations were observed between high-level PUFAs and lower incident risks of depressive disorder and anxiety disorder, which were consistent with the results for total omega-3 PUFAs and DHA. For non-LA omega-6 PUFAs, high-level PUFAs corresponded to elevated incident risks of depressive disorder, which was opposite to the results for total omega-6 PUFAs and LA (Appendix A). Notably, participants younger than 60 years old had a lower risk of depressive disorder compared with older participants, with the highest quartile of plasma omega-3 PUFA (*P* for interaction = 0.040) and DHA levels (*P* for interaction = 0.031) (Table 4). Males with high-level plasma PUFAs tended to have a lower incident risk of anxiety disorder (*P* for interaction = 0.038) (Table 4). The results demonstrated no significant differences in analyses stratified by the INFLA score and MetS (Table 5). A sensitivity analysis was performed on the above associations, which showed the robustness of our results (Appendix A).

A total of 24,003 participants with data on relevant psychiatric symptoms were further analyzed in the study to investigate the associations between plasma levels of five PUFA measures and 19 symptoms. Significant associations were observed, especially for omega-3 PUFAs and DHA. Overall, 16 out of 19 symptoms remained significantly related to the plasma levels of omega-3 PUFAs after the false discovery rate (FDR) adjustment. To be specific, the results showed that plasma levels of omega-3 PUFAs in quartile 4 (with quartile 1 as reference) were inversely related to the occurrence of adverse psychological symptoms, including the belief that one’s own life is meaningless (0.67 [95% CI, 0.56–0.80]), recent changes in the speed of moving or speaking (0.78 [95% CI, 0.63–0.97]), recent feelings of inadequacy 0.84 [95% CI, 0.75–0.94]), and others (Figure 2 and Appendix A). Plasma levels of DHA in quartile 4 showed similar associations to those described above (Appendix A). The results demonstrated no significant differences in analyses stratified by age and sex (Appendix A).

This study further evaluated the relationships between the plasma levels of total PUFAs and brain white matter microstructures within a subgroup of 8780 participants with available neuroimaging data acquired approximately 4 years after the baseline assessment. Summary characteristics of MD and ISOVF values for white matter tracts are presented in Appendix A. In terms of the MD value, 10 out of 15 associations remained significant after the FDR correction. Specifically, we observed that high plasma levels of total PUFAs were related to lower MD values of the corticospinal tract (*β* [95% CI] = −0.141 [−0.210, −0.073]), inferior longitudinal fasciculus (−0.105 [−0.173, −0.036]), forceps minor (*β* [95% CI] = −0.109 [−0.177, −0.041]), and other tracts (Figure 3 and Appendix A). Lower ISOVF values were also observed in the corticospinal tract (−0.124 [−0.196, −0.052]) and other tracts (Figure 3 and Appendix A). There was no significant difference between subgroups stratified by age and sex (Appendix A).

## 4. Discussion

Based on a large population from the UK, this study explored the associations of plasma levels of different types of PUFAs with mental health and changes in specific brain white matter microstructures. The results demonstrated that plasma PUFA levels were related to lower risks of incident depressive disorder and anxiety disorder, as well as adverse psychological symptoms, especially the omega-3 PUFAs and DHA. Additionally, high plasma PUFA levels were linked to lower MD and ISOVF values for several white matter tracts. These findings may further complement the epidemiological evidence about the effects of plasma PUFA levels on human brain health, while their correlations with particular brain structures offer insights into the mechanisms underlying the progression of mental disorders.

There have been a substantial number of studies, both animal experiments and population-based studies, supporting the antidepressant effects of specific PUFAs, especially EPA [27,28]. A recent Mendelian randomization study focused on fatty acids reported that a higher EPA level has a protective effect on the risk of incident depressive disorder, while LA has potentially detrimental effects [29]. Another cross-sectional study based on infertile women demonstrated that moderate intake of omega-3 and omega-6 PUFAs was consistently associated with a reduced risk of depressive symptoms [14]. However, a recent review suggested that while omega-3s may have an antidepressant effect on certain subgroups, the overall benefits from RCTs remain inconsistent [28]. Findings from a systematic review of randomized trials pronounced that increasing long-chain omega-3 intake probably has little or no effect on the risk of depression symptoms during 12 months or anxiety symptoms during 6 months [30]. These mixed results from RCTs suggest that the timing of the intervention, baseline omega-3 status, or other individual factors such as age and existing diseases may influence the effectiveness of omega-3 supplementation on mental health. Besides omega-3 PUFAs, the relationship between omega-6 PUFAs and the risk of incident depressive disorder has been the subject of only a limited number of studies, and the results have been inconsistent [30,31], which may be attributed to the limited sample size and short follow-up periods. The properties of inflammation resolution, maintaining membrane stability and flexibility, and oxidative stress alleviation may account for the antidepressant effects of omega-3 PUFAs [32,33,34]. Furthermore, several studies have also investigated associations between PUFA levels and neuroimaging measures, which mainly involve the grey matter volumes of the frontal lobe, temporal lobe, hippocampus, and amygdala [35,36]. A prospective cohort study also found a significant correlation between PUFAs and white matter microstructural integrity [37]. Nevertheless, given the ongoing inconsistencies in findings regarding specific PUFAs and the rapid evolution of modern dietary habits and lifestyles, it is imperative to conduct population-based repeated and extended investigations to provide insights that can inform future long-term interventions with PUFAs for the improvement and management of mental health.

Our study produced findings similar to previous studies. In our study, plasma PUFA levels were significantly associated with lower risks of depressive disorder and anxiety disorder, as well as the occurrence of adverse psychological symptoms, especially the omega-3 fatty acids and DHA. The results from multiple experiments suggest that insufficient intake of PUFAs may induce the development of depression-like behavior in laboratory animals [8,9]. The antidepressant potential of PUFAs remains a topic of debate. This effect has been observed to occur concurrently with the restoration of the underlying glymphatic system disruption and the protection of cerebrovascular function [38]. The potential antidepressant mechanisms of PUFAs are likely multifaceted. PUFAs, particularly omega-3 fatty acids, mitigate neuroinflammatory processes such as microglial activation and astrogliosis, which are implicated in depression [3]. In addition, PUFAs increase brain-derived neurotrophic factor (BDNF) levels, promoting neuronal survival and synaptic plasticity [39]. PUFA metabolites, such as EPA and DHA, improve neuronal function and antioxidant defenses [40]. Moreover, PUFAs influence serotonin and dopamine regulation, which is key for mood stabilization [41]. Although several researches have demonstrated that higher omega-6 PUFA levels have potentially detrimental effects on the risks of depressive disorder and anxiety, their proinflammatory traits are still controversial [42]. Omega-6 PUFAs produce not only proinflammatory eicosanoids but also lipid mediators that play important roles in inflammation resolution [43]. The results from our study showed that high plasma omega-6 PUFA and LA levels were significantly associated with lower risks of depressive disorder and anxiety disorder. This finding suggested that moderate intake of omega-6 PUFAs within the recommended levels is necessary for mental health. Of note, the effect of non-LA omega-6 PUFAs on mental disorders differed from that of total omega-6 PUFAs and LA. This may be due to the dual effects of arachidonic acid (AA) and its metabolites, such as prostaglandins and leukotrienes, which contribute to chronic inflammation and potentially worsen the symptoms of depression and anxiety [44,45]. However, non-LA omega-6 PUFAs may not accurately reflect the AA level, and further research is required to elucidate the dosages of different types of PUFAs and the ratios between them in order to ascertain their long-term efficacy and health effects in relation to depressive disorder and anxiety disorder. Currently, there is an increasing focus on personalized interventions, which are based on the individual patient’s unique needs, preferences, and symptoms. The results of our analysis stratified by age demonstrated that in the context of high plasma omega-3 PUFA and DHA levels, individuals younger than 60 years old exhibited a lower risk of incident depression compared to those over 60 years old. One potential explanation is that individuals under 60 years of age may have greater neuroplasticity demands, where omega-3 deficiencies can more significantly disrupt brain function [3,39]. In the neuroimaging analysis, consistent with the neuroprotective value of PUFAs from previous studies, our study found that high plasma PUFA levels were associated with lower MD values of white matter tracts, which suggests a positive correlation between PUFAs and white matter microstructural integrity. Potential mechanisms underlying the correlation may be explained by the biophysiological roles of PUFAs in myelination and white matter development in the human brain [42]. Several studies have reported the relationships between DTI-derived white matter indices and depressive disorder [46,47]. Collectively, high plasma PUFA levels might be related to decreased depressive disorder and anxiety disorder risks via their neuroprotective effects. Further studies are required to replicate and extend these findings, as well as to identify response mediators and elucidate the mechanisms of action.

Our study has several strengths. Firstly, as it is based on a prospective cohort with a large sample size and long-term follow-up, there are a substantial number of cases for our study to estimate the association between plasma levels of PUFAs and mental disorders. Secondly, this study comprehensively assessed the effects of multiple types of PUFAs on the incident risk of mental disorders. Thirdly, the neuroimaging analyses including white matter microstructures enhanced our understanding of the neurological effects. Finally, with different model strategies to gradually control diverse potential confounders involving demographic factors, a healthy lifestyle, and others, the results from our study were robust and reliable. It should be noted that the present study is not without limitations. First, as plasma PUFA levels can vary depending on recent dietary intake and other factors, PUFA profiles of other tissues (e.g., adipose tissue) or blood cells (e.g., erythrocytes) may better reflect the long-term PUFA status. Second, due to the lack of detailed data on individual omega-3 PUFAs and omega-6 PUFAs other than DHA and LA, this study only estimated the impacts of non-DHA omega-3 PUFAs and non-LA omega-6 PUFAs, which may not be an accurate representation of the roles of other important individual PUFAs, including EPA and AA. Further research is needed to investigate the effects of individual fatty acids on mental health. Third, given that the incidences of mental disorders were ascertained through electronic health records, cases may be underreported or the diagnosis may be delayed. Fourth, despite controlling for numerous widely accepted confounders, we cannot avoid the possibility of residual or unmeasured confounding effects. For instance, although we adjusted for a healthy diet related to brain health in our analysis model, the effects of other medications were not discussed in this study. Future investigations with detailed data on medication use are warranted. Fifth, the mere association disclosed by regression modeling did not imply causality, and substantial evidence is still needed to support the biological plausibility of our findings. Finally, in this population-based cohort from the UK Biobank, participants were predominantly of European Caucasian descent, with nearly 97% of the database identifying as ‘White’, and individuals of non-European ancestry were underrepresented. Therefore, the results may have limited applicability to other ethnic groups, and further studies are required to confirm them in other ethnic groups.

## 5. Conclusions

In conclusion, our findings demonstrated that plasma levels of PUFAs, especially omega-3 fatty acids and DHA, were inversely associated with the risks of depressive disorder and anxiety disorder, as well as the occurrence of adverse psychological symptoms. Further neuroimaging analyses indicated that high plasma PUFA levels were associated with lower MD and ISOVF values for several white matter tracts, which indicated a positive correlation with the integrity of white matter. This study provides population-based evidence to support the value of interventions targeting PUFAs (specifically omega-3 PUFAs) in the prevention and management of mental disorders.

## Figures and Tables

**Figure 1 nutrients-16-04065-f001:**
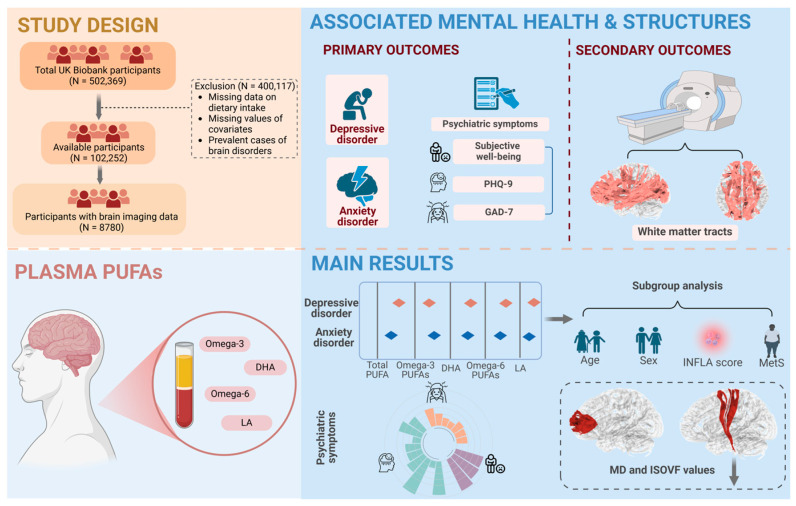
Study workflow. Our study extracted data from 102,252 residents in the UK Biobank with longitudinal outcomes, including depressive and anxiety disorders. The related psychiatric symptoms were also considered as primary outcomes. Brain white matter microstructures were derived as secondary outcomes. The study population, exclusions, and missing data are also presented. Cox proportional hazard regression models were utilized to assess the associations between plasma PUFA levels and mental disorders. Hazard ratios (HRs) and the corresponding 95% confidence intervals (CIs) were calculated in our analyses. Subgroup analyses were conducted and were mainly stratified by age (<60/≥60), sex (male/female), INFLA score (low/high), and MetS (yes/no). Further analyses were conducted to estimate the relationships of plasma PUFAs with symptoms and white matter microstructures by logistic regression models and multiple linear regression models, respectively.

**Figure 2 nutrients-16-04065-f002:**
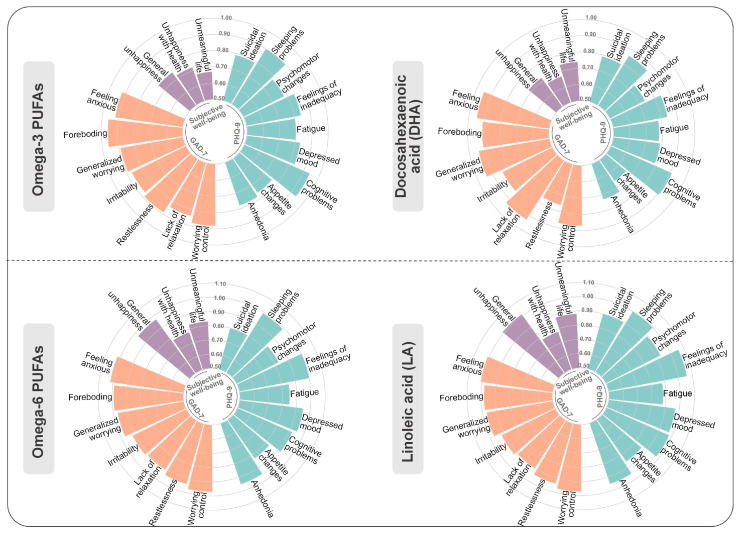
Associations between plasma PUFA levels and psychiatric symptoms. The relationships of omega-3 PUFAs, DHA, omega-6 PUFAs, and LA, with the 19 symptoms were estimated among 24,003 participants with available information utilizing logistic regression models. Data are presented as the ORs. Significance was determined through the FDR-corrected *p* value.

**Figure 3 nutrients-16-04065-f003:**
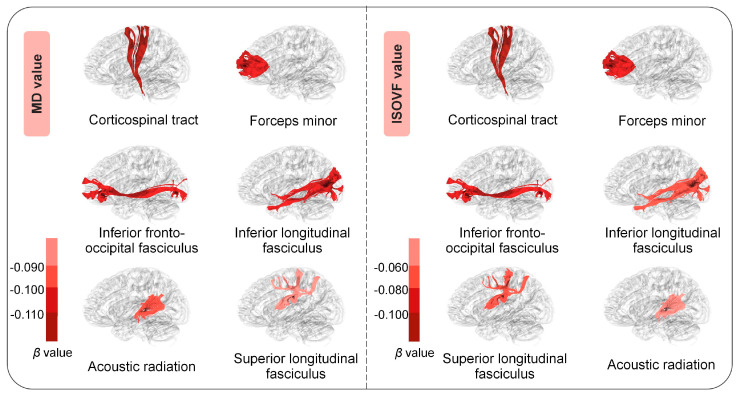
Associations between plasma PUFA levels and white matter tracts. The associations between plasma PUFA levels and the MD and ISOVF values of white matter tracts were estimated among 8780 participants with available neuroimaging data utilizing multiple linear regression models. Data are presented as *β*s. Significance was determined through the FDR-corrected *p* value.

**Table 1 nutrients-16-04065-t001:** Baseline characteristics of participants grouped by plasma levels of total PUFAs.

Characteristic	All Participants (N = 102,252)	Total PUFA Quartiles				*p* Value
		Q1 ^a^ (N = 25,566)	Q2 (N = 25,563)	Q3 (N = 25,562)	Q4 (N = 25,561)	
Age, years						<0.001
<60	57,863 (56.6%)	15,262 (59.7%)	15,138 (59.2%)	14,344 (56.1%)	13,119 (51.3%)	
≥60	44,389 (43.4%)	10,304 (40.3%)	10,425 (40.8%)	11,218 (43.9%)	12,442 (48.7%)	
Sex						<0.001
Female	54,440 (53.2%)	10,218 (40.0%)	12,939 (50.6%)	14,668 (57.4%)	16,615 (65.0%)	
Male	47,812 (46.8%)	15,348 (60.0%)	12,624 (49.4%)	10,894 (42.6%)	8946 (35.0%)	
IMD ^b^						<0.001
Low	51,102 (50.0%)	12,030 (47.1%)	12,766 (49.9%)	13,139 (51.4%)	13,167 (51.5%)	
High	51,150 (50.0%)	13,536 (52.9%)	12,797 (50.1%)	12,423 (48.6%)	12,394 (48.5%)	
WHR ^c^						<0.001
Ideal	52,390 (51.3%)	12,182 (47.8%)	13,420 (52.6%)	13,655 (53.5%)	13,133 (51.5%)	
Poor	49,661 (48.7%)	13,323 (52.2%)	12,094 (47.4%)	11,864 (46.5%)	12,380 (48.5%)	
Lifestyle						
No current smoking	91,768 (90.2%)	22501 (88.5%)	22921 (90.1%)	23114 (90.8%)	23232 (91.4%)	<0.001
Regular physical activity ^d^	45,807 (55.2%)	11,254 (53.8%)	11,576 (55.5%)	11,579 (55.7%)	11,398 (55.8%)	<0.001
Healthy sleep pattern ^e^	55,776 (54.5%)	13,987 (54.7%)	14,106 (55.2%)	13,930 (54.5%)	13,753 (53.8%)	0.017
Healthy diet ^f^	39,164 (38.3%)	8978 (35.1%)	9491 (37.1%)	10,039 (39.3%)	10,656 (41.7%)	<0.001
Blood biomarkers						
SBP, mmHg	137.86 (18.55)	135.12 (18.23)	137.03 (18.49)	138.63 (18.46)	140.65 (18.57)	<0.001
DBP, mmHg	82.15 (10.12)	80.69 (10.17)	81.92 (10.12)	82.58 (10.04)	83.41 (9.98)	<0.001
Blood glucose, mmol/L	5.11 (1.19)	5.18 (1.43)	5.08 (1.14)	5.07 (1.05)	5.10 (1.09)	<0.001

Abbreviations: PUFAs, polyunsaturated fatty acids; IMD, index of multiple deprivation; WHR, waist-to-hip ratio; SBP, systolic blood pressure; DBP, diastolic blood pressure. ^a^ Plasma PUFA levels were divided into quartiles. ^b^ IMD scores offer a more complex and detailed view of deprivation based on more factors than the Townsend index. Domains for the IMD calculation include the crime score (England and Scotland), community safety score (Wales), education score (all), employment score (all), health score (all), housing score (all), income score (all), living environment score (England), access to services score (Scotland and Wales), and physical environment score (Wales). ^c^ WHR was calculated as the waist circumference (centimeters) divided by the hip circumference (centimeters) and categorized into ideal (<0.9 for men and <0.85 for women) and poor (≥0.9 for men and ≥0.85 for women). ^d^ Regular physical activity: ≥150 min of moderate-intensity exercise per week, ≥75 min of vigorous-intensity exercise per week, or a combination of both. ^e^ A healthy sleep pattern was measured by 5 dimensions of sleep behaviors: early chronotype, sleep 7–8 h/day, never/rarely or sometimes insomnia, no self-reported snoring, and never/rarely or sometimes excessive daytime sleepiness. It was categorized into yes (≥4 healthy components) and no. ^f^ A healthy diet was defined based on at least four of seven aspects of healthy diet components: fruits ≥ 3 servings/day; vegetables ≥ 3 servings/day; fish ≥ 2 servings/week; processed meats ≤ 1 serving/week; unprocessed red meats ≤ 1.5 servings/week; whole grains ≥ 3 servings/day; and refined grains ≤ 1.5 servings/day.

**Table 2 nutrients-16-04065-t002:** Associations between quartiles of plasma levels of the five PUFA measures and the incident risks of depressive disorder and anxiety disorder.

Mental Disorders ^a^	Per Quartile, mmol/L				Per IQR
	Q1	Q2	Q3	Q4	
Total PUFAs ^b^					
Depressive disorder					
Model 1 ^a^	Reference	0.85 (0.77, 0.93)	0.77 (0.70, 0.85)	0.77 (0.70, 0.84)	0.87 (0.84, 0.92)
Model 2 ^b^	Reference	0.85 (0.77, 0.95)	0.76 (0.68, 0.85)	0.78 (0.70, 0.87)	0.89 (0.84, 0.93)
Model 3 ^c^	Reference	0.85 (0.76, 0.96)	0.77 (0.69, 0.87)	0.80 (0.71, 0.90)	0.90 (0.85, 0.95)
Anxiety disorder					
Model 1	Reference	0.94 (0.85, 1.03)	0.85 (0.77, 0.93)	0.82 (0.74, 0.90)	0.91 (0.87, 0.95)
Model 2	Reference	0.99 (0.88, 1.10)	0.88 (0.79, 0.99)	0.82 (0.73, 0.92)	0.90 (0.86, 0.95)
Model 3	Reference	0.97 (0.86, 1.09)	0.89 (0.79, 1.01)	0.83 (0.74, 0.94)	0.91 (0.86, 0.97)
Omega-3 PUFAs ^c^					
Depressive disorder					
Model 1	Reference	0.86 (0.79, 0.95)	0.85 (0.78, 0.94)	0.80 (0.73, 0.88)	0.92 (0.88, 0.96)
Model 2	Reference	0.90 (0.81, 1.01)	0.88 (0.79, 0.99)	0.86 (0.77, 0.97)	0.95 (0.91, 1.00)
Model 3	Reference	0.89 (0.79, 1.01)	0.86 (0.76, 0.97)	0.86 (0.76, 0.97)	0.96 (0.91, 1.01)
Anxiety disorder					
Model 1	Reference	0.91 (0.82, 1.00)	0.85 (0.77, 0.93)	0.84 (0.76, 0.92)	0.93 (0.89, 0.97)
Model 2	Reference	0.90 (0.81, 1.00)	0.83 (0.74, 0.93)	0.82 (0.74, 0.92)	0.93 (0.89, 0.98)
Model 3	Reference	0.88 (0.78, 1.00)	0.82 (0.72, 0.92)	0.83 (0.73, 0.94)	0.95 (0.90, 1.00)
DHA ^d^					
Depressive disorder					
Model 1	Reference	0.86 (0.78, 0.94)	0.82 (0.74, 0.90)	0.73 (0.66, 0.80)	0.87 (0.84, 0.91)
Model 2	Reference	0.86 (0.77, 0.96)	0.86 (0.77, 0.96)	0.79 (0.71, 0.89)	0.92 (0.87, 0.96)
Model 3	Reference	0.89 (0.79, 1.00)	0.84 (0.75, 0.95)	0.80 (0.71, 0.91)	0.92 (0.87, 0.97)
Anxiety disorder					
Model 1	Reference	0.90 (0.81, 0.99)	0.83 (0.76, 0.92)	0.76 (0.69, 0.84)	0.90 (0.86, 0.94)
Model 2	Reference	0.91 (0.81, 1.01)	0.86 (0.77, 0.96)	0.79 (0.70, 0.88)	0.91 (0.87, 0.96)
Model 3	Reference	0.90 (0.80, 1.02)	0.88 (0.78, 1.00)	0.80 (0.71, 0.91)	0.93 (0.88, 0.98)
Omega-6 PUFAs ^e^					
Depressive disorder					
Model 1	Reference	0.86 (0.78, 0.94)	0.82 (0.74, 0.90)	0.77 (0.70, 0.84)	0.88 (0.84, 0.92)
Model 2	Reference	0.87 (0.78, 0.97)	0.80 (0.72, 0.90)	0.77 (0.69, 0.86)	0.88 (0.84, 0.93)
Model 3	Reference	0.86 (0.77, 0.97)	0.84 (0.74, 0.94)	0.79 (0.70, 0.89)	0.90 (0.85, 0.95)
Anxiety disorder					
Model 1	Reference	1.00 (0.91, 1.10)	0.90 (0.81, 0.99)	0.84 (0.76, 0.93)	0.91 (0.87, 0.96)
Model 2	Reference	1.03 (0.93, 1.15)	0.93 (0.83, 1.04)	0.85 (0.75, 0.95)	0.91 (0.86, 0.96)
Model 3	Reference	1.00 (0.89, 1.13)	0.94 (0.83, 1.06)	0.84 (0.75, 0.96)	0.92 (0.86, 0.97)
LA ^f^					
Depressive disorder					
Model 1	Reference	0.83 (0.75, 0.91)	0.81 (0.73, 0.89)	0.75 (0.68, 0.82)	0.86 (0.83, 0.90)
Model 2	Reference	0.81 (0.73, 0.90)	0.78 (0.70, 0.87)	0.75 (0.67, 0.84)	0.87 (0.82, 0.91)
Model 3	Reference	0.80 (0.71, 0.91)	0.83 (0.74, 0.94)	0.77 (0.69, 0.87)	0.88 (0.84, 0.94)
Anxiety disorder					
Model 1	Reference	0.95 (0.86, 1.04)	0.88 (0.80, 0.97)	0.83 (0.76, 0.92)	0.91 (0.87, 0.95)
Model 2	Reference	0.98 (0.88, 1.09)	0.92 (0.83, 1.03)	0.83 (0.74, 0.93)	0.91 (0.86, 0.95)
Model 3	Reference	0.96 (0.86, 1.09)	0.94 (0.84, 1.06)	0.84 (0.75, 0.95)	0.92 (0.87, 0.97)

Abbreviations: PUFAs, polyunsaturated fatty acids; DHA, docosahexaenoic acid; LA, linoleic acid. ^a^ Three model strategies were adopted for the analyses: Model 1 was adjusted for baseline characteristics, including age, sex, WHR, and IMD; Model 2 was additionally adjusted for healthy lifestyles (never drinking, a healthy sleep pattern, healthy diet, and regular physical activity); Model 3 was further additionally adjusted for systolic blood pressure, diastolic blood pressure, and blood glucose levels. ^b^ The plasma level of total PUFAs was divided into quartiles: Q1 (<4.43 mmol/L), Q2 (4.43 to 4.93 mmol/L), Q3 (4.93 to 5.47 mmol/L), and Q4 (>5.47 mmol/L). ^c^ The plasma level of omega-3 PUFAs was divided into quartiles: Q1 (<0.37 mmol/L), Q2 (0.37 to 0.49 mmol/L), Q3 (0.49 to 0.64 mmol/L), and Q4 (>0.64 mmol/L). ^d^ The plasma level of DHA was divided into quartiles: Q1 (<0.18 mmol/L), Q2 (0.18 to 0.22 mmol/L), Q3 (0.22 to 0.28 mmol/L), and Q4 (>0.28 mmol/L). ^e^ The plasma level of omega-6 PUFAs was divided into quartiles: Q1 (<3.99 mmol/L), Q2 (3.99 to 4.41 mmol/L), Q3 (4.41 to 4.87 mmol/L), and Q4 (>4.87 mmol/L). ^f^ The plasma level of LA was divided into quartiles: Q1 (<2.95 mmol/L), Q2 (2.95 to 3.37 mmol/L), Q3 (3.37 to 3.83 mmol/L), and Q4 (>3.83 mmol/L).

**Table 3 nutrients-16-04065-t003:** Associations between the quintiles of plasma levels of seven PUFA measures and the incident risks of depressive disorder and anxiety disorder.

Mental Disorders ^a^	Per Quintile, mmol/L				
	Quintile 1	Quintile 2	Quintile 3	Quintile 4	Quintile 5
Total PUFAs ^b^					
Depressive disorder					
Model 1	Reference	0.82 (0.73, 0.90)	0.79 (0.71, 0.88)	0.71 (0.64, 0.79)	0.77 (0.69, 0.85)
Model 2	Reference	0.79 (0.70, 0.90)	0.81 (0.72, 0.91)	0.67 (0.59, 0.76)	0.79 (0.70, 0.89)
Model 3	Reference	0.79 (0.69, 0.90)	0.85 (0.75, 0.97)	0.68 (0.59, 0.78)	0.81 (0.71, 0.93)
Anxiety disorder					
Model 1	Reference	0.94 (0.84, 1.04)	0.89 (0.80, 1.00)	0.80 (0.71, 0.89)	0.82 (0.74, 0.92)
Model 2	Reference	0.94 (0.83, 1.06)	0.95 (0.84, 1.07)	0.80 (0.71, 0.91)	0.83 (0.73, 0.94)
Model 3	Reference	0.91 (0.79, 1.04)	0.96 (0.84, 1.09)	0.78 (0.68, 0.90)	0.84 (0.73, 0.96)
Omega-3 PUFAs ^c^					
Depressive disorder					
Model 1	Reference	0.88 (0.79, 0.97)	0.85 (0.76, 0.94)	0.78 (0.70, 0.87)	0.79 (0.71, 0.88)
Model 2	Reference	0.91 (0.80, 1.02)	0.90 (0.80, 1.02)	0.81 (0.71, 0.92)	0.86 (0.76, 0.98)
Model 3	Reference	0.90 (0.79, 1.03)	0.90 (0.79, 1.03)	0.79 (0.69, 0.91)	0.86 (0.75, 0.99)
Anxiety disorder					
Model 1	Reference	0.91 (0.82, 1.02)	0.90 (0.81, 1.00)	0.81 (0.72, 0.90)	0.82 (0.73, 0.91)
Model 2	Reference	0.89 (0.79, 1.01)	0.93 (0.82, 1.05)	0.81 (0.71, 0.91)	0.80 (0.71, 0.91)
Model 3	Reference	0.88 (0.77, 1.01)	0.94 (0.82, 1.07)	0.81 (0.70, 0.93)	0.82 (0.71, 0.94)
DHA ^d^					
Depressive disorder					
Model 1	Reference	0.93 (0.84, 1.03)	0.80 (0.72, 0.89)	0.78 (0.70, 0.87)	0.72 (0.65, 0.81)
Model 2	Reference	0.94 (0.84, 1.06)	0.84 (0.74, 0.95)	0.86 (0.76, 0.97)	0.79 (0.70, 0.90)
Model 3	Reference	0.97 (0.85, 1.10)	0.83 (0.72, 0.94)	0.86 (0.75, 0.99)	0.80 (0.69, 0.92)
Anxiety disorder					
Model 1	Reference	0.96 (0.87, 1.07)	0.81 (0.73, 0.91)	0.85 (0.76, 0.95)	0.74 (0.66, 0.83)
Model 2	Reference	1.02 (0.91, 1.16)	0.86 (0.76, 0.97)	0.89 (0.78, 1.01)	0.79 (0.69, 0.90)
Model 3	Reference	1.05 (0.92, 1.20)	0.86 (0.75, 0.99)	0.92 (0.80, 1.06)	0.82 (0.71, 0.95)
Omega-6 PUFAs ^e^					
Depressive disorder					
Model 1	Reference	0.80 (0.72, 0.89)	0.77 (0.70, 0.86)	0.74 (0.67, 0.82)	0.76 (0.69, 0.84)
Model 2	Reference	0.79 (0.70, 0.89)	0.76 (0.67, 0.85)	0.70 (0.62, 0.79)	0.77 (0.68, 0.87)
Model 3	Reference	0.78 (0.68, 0.89)	0.78 (0.68, 0.88)	0.71 (0.62, 0.82)	0.80 (0.70, 0.91)
Anxiety disorder					
Model 1	Reference	0.90 (0.81, 1.01)	0.87 (0.78, 0.97)	0.81 (0.73, 0.91)	0.83 (0.74, 0.92)
Model 2	Reference	0.89 (0.78, 1.00)	0.90 (0.80, 1.02)	0.82 (0.72, 0.93)	0.81 (0.72, 0.92)
Model 3	Reference	0.85 (0.74, 0.97)	0.89 (0.78, 1.02)	0.82 (0.72, 0.94)	0.82 (0.71, 0.94)
LA ^f^					
Depressive disorder					
Model 1	Reference	0.83 (0.75, 0.92)	0.76 (0.68, 0.84)	0.75 (0.68, 0.83)	0.73 (0.65, 0.81)
Model 2	Reference	0.84 (0.75, 0.95)	0.72 (0.64, 0.81)	0.75 (0.66, 0.85)	0.73 (0.65, 0.83)
Model 3	Reference	0.84 (0.74, 0.95)	0.73 (0.63, 0.83)	0.78 (0.69, 0.89)	0.76 (0.67, 0.87)
Anxiety disorder					
Model 1	Reference	0.99 (0.89, 1.1)	0.86 (0.77, 0.96)	0.85 (0.76, 0.95)	0.83 (0.74, 0.92)
Model 2	Reference	0.98 (0.87, 1.11)	0.89 (0.79, 1.01)	0.87 (0.77, 0.98)	0.82 (0.73, 0.93)
Model 3	Reference	0.94 (0.82, 1.07)	0.87 (0.76, 1.00)	0.88 (0.77, 1.01)	0.83 (0.72, 0.95)

Abbreviations: PUFAs, polyunsaturated fatty acids; DHA, docosahexaenoic acid; LA, linoleic acid. ^a^ Three model strategies were adopted for the analyses: Model 1 was adjusted for the baseline characteristics, including age, sex, WHR, and IMD; Model 2 was additionally adjusted for healthy lifestyles (never drinking, a healthy sleep pattern, healthy diet, and regular physical activity); Model 3 was further additionally adjusted for systolic blood pressure, diastolic blood pressure, and blood glucose levels. ^b^ The plasma level of total PUFAs was divided into quintiles: Q1 (<4.31 mmol/L), Q2 (4.31 to 4.74 mmol/L), Q3 (4.74 to 5.13 mmol/L), Q4 (5.13 to 5.61 mmol/L), and Q5 (>5.61 mmol/L). ^c^ The plasma level of omega-3 PUFAs was divided into quintiles: Q1 (<0.35 mmol/L), Q2 (0.35 to 0.45 mmol/L), Q3 (0.45 to 0.55 mmol/L), Q4 (0.55 to 0.68 mmol/L), and Q5 (>0.68 mmol/L). ^d^ The plasma level of DHA was divided into quintiles: Q1 (<0.17 mmol/L), Q2 (0.17 to 0.21 mmol/L), Q3 (0.21 to 0.24 mmol/L), Q4 (0.24 to 0.29 mmol/L), and Q5 (>0.29 mmol/L). ^e^ The plasma level of omega-6 PUFAs was divided into quintiles: Q1 (<3.89 mmol/L), Q2 (3.89 to 4.25 mmol/L), Q3 (4.25 to 4.58 mmol/L), Q4 (4.58 to 4.99 mmol/L), and Q5 (>4.99 mmol/L). ^f^ The plasma level of LA was divided into quintiles: Q1 (<2.85 mmol/L), Q2 (2.85 to 3.21 mmol/L), Q3 (3.21 to 3.54 mmol/L), Q4 (3.54 to 3.95 mmol/L), and Q5 (>3.95 mmol/L).

**Table 4 nutrients-16-04065-t004:** Associations between plasma PUFA levels and mental disorders stratified by age and sex.

Plasma PUFAs	Mental Disorders	Age ^a^		*P* for Interaction ^b^	Sex		*P* for Interaction
Total PUFAs	Depressive disorder	<60	0.75 (0.63, 0.88)	0.269	Male	0.81 (0.67, 0.97)	0.352
		≥60	0.85 (0.72, 1.02)		Female	0.82 (0.70, 0.97)	
	Anxiety disorder	<60	0.88 (0.74, 1.05)	0.348	Male	0.75 (0.61, 0.92)	0.038
		≥60	0.78 (0.65, 0.92)		Female	0.91 (0.78, 1.08)	
Omega-3 PUFAs	Depressive disorder	<60	0.80 (0.68, 0.94)	0.040	Male	0.90 (0.75, 1.09)	0.436
		≥60	0.97 (0.80, 1.18)		Female	0.83 (0.71, 0.98)	
	Anxiety disorder	<60	0.90 (0.76, 1.06)	0.984	Male	0.72 (0.59, 0.87)	0.072
		≥60	0.80 (0.67, 0.96)		Female	0.93 (0.79, 1.08)	
DHA	Depressive disorder	<60	0.73 (0.62, 0.87)	0.031	Male	0.74 (0.60, 0.92)	0.667
		≥60	0.90 (0.75, 1.09)		Female	0.82 (0.69, 0.96)	
	Anxiety disorder	<60	0.80 (0.67, 0.96)	0.556	Male	0.70 (0.56, 0.87)	0.235
		≥60	0.83 (0.69, 0.99)		Female	0.86 (0.73, 1.01)	
Omega-6 PUFAs	Depressive disorder	<60	0.76 (0.64, 0.89)	0.615	Male	0.73 (0.60, 0.88)	0.186
		≥60	0.82 (0.69, 0.98)		Female	0.86 (0.73, 1.01)	
	Anxiety disorder	<60	0.88 (0.74, 1.05)	0.496	Male	0.75 (0.61, 0.93)	0.069
		≥60	0.81 (0.68, 0.96)		Female	0.93 (0.79, 1.09)	
LA	Depressive disorder	<60	0.74 (0.63, 0.87)	0.566	Male	0.68 (0.56, 0.83)	0.103
		≥60	0.81 (0.68, 0.97)		Female	0.84 (0.72, 0.98)	
	Anxiety disorder	<60	0.84 (0.70, 1.00)	0.763	Male	0.76 (0.62, 0.93)	0.136
		≥60	0.85 (0.71, 1.01)		Female	0.90 (0.77, 1.06)	

Abbreviation: DHA, docosahexaenoic acid; LA, linoleic acid. ^a^ Effect values are presented as hazard ratios with corresponding 95% CIs. ^b^ *P* for interaction was assessed based on a multiplicative scale by the addition of a product term to the model for testing the statistical significance of heterogeneity.

**Table 5 nutrients-16-04065-t005:** Associations between plasma PUFA levels and mental disorders stratified by the INFLA score and MetS.

Plasma PUFAs	Mental Disorders	INFLA Score ^a^		*P* for Interaction ^b^	MetS		*P* for Interaction
Total PUFAs	Depressive disorder	Low	0.82 (0.69, 0.97)	0.670	Yes	0.71 (0.59, 0.85)	0.123
		High	0.79 (0.67, 0.93)		No	0.88 (0.75, 1.04)	
	Anxiety disorder	Low	0.87 (0.73, 1.04)	0.757	Yes	0.74 (0.61, 0.89)	0.073
		High	0.79 (0.67, 0.94)		No	0.91 (0.77, 1.06)	
Omega-3 PUFAs	Depressive disorder	Low	0.91 (0.77, 1.09)	0.371	Yes	0.77 (0.64, 0.93)	0.107
		High	0.83 (0.70, 0.98)		No	0.92 (0.78, 1.07)	
	Anxiety disorder	Low	0.79 (0.66, 0.93)	0.256	Yes	0.76 (0.62, 0.93)	0.319
		High	0.89 (0.75, 1.05)		No	0.88 (0.75, 1.02)	
DHA	Depressive disorder	Low	0.83 (0.70, 0.98)	0.331	Yes	0.71 (0.58, 0.87)	0.090
		High	0.78 (0.66, 0.94)		No	0.89 (0.76, 1.05)	
	Anxiety disorder	Low	0.70 (0.59, 0.84)	0.074	Yes	0.69 (0.56, 0.85)	0.170
		High	0.94 (0.79, 1.12)		No	0.88 (0.75, 1.03)	
Omega-6 PUFAs	Depressive disorder	Low	0.77 (0.65, 0.91)	0.424	Yes	0.74 (0.62, 0.89)	0.227
		High	0.81 (0.69, 0.96)		No	0.84 (0.71, 0.98)	
	Anxiety disorder	Low	0.89 (0.74, 1.06)	0.644	Yes	0.78 (0.65, 0.95)	0.127
		High	0.80 (0.67, 0.95)		No	0.89 (0.76, 1.05)	
LA	Depressive disorder	Low	0.74 (0.62, 0.87)	0.256	Yes	0.75 (0.62, 0.89)	0.464
		High	0.81 (0.69, 0.96)		No	0.80 (0.68, 0.94)	
	Anxiety disorder	Low	0.87 (0.73, 1.04)	0.942	Yes	0.84 (0.69, 1.01)	0.473
		High	0.82 (0.69, 0.97)		No	0.85 (0.72, 1.00)	

Abbreviation: MetS, metabolic syndrome; DHA, docosahexaenoic acid; LA, linoleic acid. ^a^ Effect values are presented as hazard ratios with corresponding 95% CIs. ^b^ *P* for interaction was assessed based on a multiplicative scale by the addition of a product term to the model for testing the statistical significance of heterogeneity.

## Data Availability

All the data for this study will be made available upon reasonable request to the corresponding authors.

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
