# Peer review of "Plasma Polyunsaturated Fatty Acid Levels and Mental Health in Middle-Aged and Elderly Adults"

_nutrients, 2024, doi:10.3390/nu16234065_

Round 1
Reviewer 1 Report
Comments and Suggestions for Authors
This ms examined data from the UK Biobank study, exploring the relationship between high and low PUFA on various parameters related to mental health and white matter microstructures.
There are several matters here for the authors to consider.
1. There have been many papers from the UK Biobank study. What is novel about this ms?
2. The Discussion should include mention of the strengths of the study.
3. What was the rationale for measuring only 2 conditions (high and low PUFA)? Why not look at dividing plasma PUFA into quintiles?
Reviewer 2 Report
Comments and Suggestions for Authors
The study presents an interesting investigation of the relationship between plasma polyunsaturated fatty acid (PUFA) levels and mental health in a UK adult population. The study primarily concentrates on the summarised data of PUFAs, including n-3 and n-6 PUFAs. However, the methodology section lacks sufficient detail regarding the exact number and types of fatty acids included in these summarised values. Please provide the aforementioned information.
Furthermore, it is unclear why EPA was not included in this study, given that it plays a pivotal role in several mental disorders, such as depression, as mentioned in the introduction.
The two most important PUFAs in the brain are DHA and AA, yet only total n-6 PUFAs were included, without the parent fatty acid AA. Could you please explain why this fatty acid was omitted?If you have AA levels from this study group, did low AA levels in adults increase the incident risk of the investigated mental disorders, like n-6 PUFAs or did it have opposite effect?
If there are no information of these fatty acid data in the original database then please include this fact in the Methods section and also in the limitations in the Discussion part.
Some minor remarks:
Figure 1: on the picture correct second(ary) outcomes
Table 1: it look s bit strange that all three groups have blood glucose 5.1 mmol/l and the difference is significant. Can you give these values with two decimals?
Figure 4: plasma PUFA levels: delete ‘s’ from PUFAs PUFAs is the plural for of polyunsaturated fatty acids, but if you use for example the word level after this, you have to put it in the singular form. Please revise this mistake in the whole manuscript.
Line 66: PUFAs (add ‘s’)
line 277 correct omega-6 (change ‘e/a’ order)
line 284: significant association (delete ‘ly’)
line 400-401: use the abbreviation PUFAs
